

# Acute effects of contract-relax proprioceptive neuromuscular facilitation stretching of hip abductors and adductors on dynamic balance

Rafał Szafraniec[1], Krystyna Chromik[1], Amanda Poborska[2] and Adam Kawczyński[1]

[1] Faculty of Sports Science, University School of Physical Education, Wrocław, Poland
[2] Faculty of Natural Sciences and Technology, Karkonosze College, Jelenia Góra, Poland

## ABSTRACT

**Background**. Balance control has been shown to play a fundamental role both in everyday life and many athletic activities. An important component of balance control is the somatosensory information gained from muscle spindles and Golgi tendon organs. The changes in the muscle-tendon unit stiffness could alter the ability to detect and respond promptly to changes of an unstable environment. One of the procedures affecting muscle stiffness is stretching, and contract-relax PNF stretching (CRS) is considered as one of the safest and most effective techniques. So far, there are no studies on the impact of CRS of hip adductor and abductor muscles on body balance. These muscle groups are responsible for maintaining mediolateral balance which is of particular interest, since it is more affected by ageing and disease and since its deterioration has been associated with an increased risk of falling. In light of the above, the aim of the present study was to investigate the effects of a single dose of contract-relax proprioceptive neuromuscular facilitation stretching of hip adductors and abductors on mediolateral dynamic balance.

**Methods**. The study involved 45 healthy individuals (age 19–23 years) assigned to the intervention group (IG) or the control group (CG). Balance testing was carried out before (Pre) and immediately after CRS in the intervention group or after 5-minute rest in the control group (Post). There were performed three repetitions of the CRS targeting the adductor and abductor muscles of the hip.

**Results**. Statistically significant differences between Pre and Post condition were observed only in the intervention group. The values of all measured variables defining the body's dynamic balance were significantly lower immediately after the applied CRS, which indicates an improved body balance: Global Index ($p = 0.0001$), total area of sway ($p = 0.0001$), external area of sway ($p = 0.00004$), external time ($p = 0.0004$) and reaction time ($p = 0.0003$).

**Conclusions**. A single dose of contract-relax proprioceptive neuromuscular facilitation stretching of the hip adductor and abductor muscles improved mediolateral dynamic balance.

Corresponding author
Rafał Szafraniec,
rafal.szafraniec@awf.wroc.pl

## INTRODUCTION

Balance control has been shown to play a fundamental role both in everyday life and many athletic activities (*Behm et al., 2015*). Postural control is the output of muscle activity that counteracts the destabilizing effects of gravity and inertia in static and dynamic conditions (*Duarte & Sternad, 2008*). The ability to maintain balance is based on multiple components that coordinate sensory input and motor output (*Li et al., 2018*; *Dos Santos et al., 2017*; *Stins, Roerdink & Beek, 2011*; *Manor et al., 2010*). Due to differences in postural control schemes, the convention has classified balance as either static or dynamic. Static balance is understood as the ability to maintain the body in a fixed upright position with minimal sway over the base of support (*Yamagata et al., 2017*). In turn, dynamic balance is the ability to maintain postural stability when the body performs a motor activity with a moving base of support (*Gürkan et al., 2016*). An important component of balance control is the somatosensory information gained from muscle spindles and Golgi tendon organs. Changes in muscle–tendon stiffness could alter the ability to detect and respond promptly to changes of an unstable environment (*Herda et al., 2011*). On the other hand, flexibility of balance control system is achieved by reweighting sensory inputs based on reliability to balance control in a given situation (*Welch & Ting, 2009*).

One of the procedures affecting muscle stiffness is stretching (*Taniguchi et al., 2015*). Three common stretching techniques include static stretching, dynamic stretching and proprioceptive neuromuscular facilitation (PNF) (*Behm et al., 2015*). Two PNF techniques are the most popular: the contract-relax method and the contract-relax-antagonist-contract method (CRAC) (*Hindle et al., 2012*). Contract-relax stretching (CRS) is considered as one of the safest and most effective stretching techniques. The mechanics of CRS are based on the reflex action of the Golgi tendon organs after isometric tension is applied. Then the motoneuron fields of the activated muscle are attenuated to reduce muscle tension (*Hindle et al., 2012*). In balance control, changes in a muscle tone may mitigate the effects of a perturbation by changing the mechanical response of the body to perturbation (*Welch & Ting, 2009*).

Most studies examined the effects of static stretching on dynamic balance performance but the results are not unequivocal. Some authors reported that static stretching enhances dynamic balance (*Nelson et al., 2012*; *Handrakis et al., 2010*; *Costa et al., 2009*). On the contrary, some studies confirmed the negative effect of static stretching on the body balance (*Chatzopoulos et al., 2014*; *Han et al., 2014*; *Lima et al., 2014*; *Nagano et al., 2006*; *Behm et al., 2004*).

Data examining the effects of PNF stretching on balance control are scarce. *Ghram, Damak & Costa (2017)* stated that CRS of the quadriceps, hamstrings, anterior tibialis, and calf muscles impaired static balance control in healthy men. In contrast, *Ryan, Rossi & Lopez (2010)* found that CRAC of the hamstrings, plantar and hip flexor muscles improved mediolateral (ML) balance. The authors observed improved stability in the ML plane, despite the fact that the flexor muscles of the lower limb were stretched, i.e., the muscles controlling the balance in anteroposterior plane. The ML balance, however, is controlled by hip adductor and abductor muscles (*Horak, 2006*). We therefore suspect that the

**Table 1  Anthropometric characteristics of the participants (mean ± SD).**

|  | IG | | CG | |
|---|---|---|---|---|
|  | Female ($n = 20$) | Male ($n = 11$) | Female ($n = 9$) | Male ($n = 5$) |
| Age (years) | 20.7 ± 1.2 | 21.6 ± 1.7 | 20.2 ± 1.1 | 21.2 ± 1.3 |
| Body height (cm) | 169.4 ± 6.9 | 178 ± 4.9 | 167.8 ± 5.7 | 176.4 ± 4.1 |
| Body mass (kg) | 65.8 ± 9.6 | 81 ± 10.8 | 63.9 ± 7.8 | 78.6 ± 8.9 |
| BMI (kg/m$^2$) | 23 ± 3.2 | 25.5 ± 2.7 | 22.9 ± 2.8 | 25.3 ± 2.4 |
| Fat (%) | 33.3 ± 6.0 | 20.1 ± 5.1 | 31.5 ± 5.1 | 19.8 ± 4.8 |

stretching of these muscle groups can more significantly improve the ML balance than the stretching of hamstrings or plantar and hip flexors. *Ryan, Rossi & Lopez (2010)* suspect that the improvement in ML stability may be because of neurological facilitation from the contract–relax portion, or irradiation overflow from the antagonist-contract phase. To determine which mechanism is more likely, we decided to apply in our research the CRS method, in which there is no the antagonist-contract phase.

Confirmation of the hypothesis that CRS of hip adductors and abductors improves ML balance would have an application value, because efficient body balance is important not only in sports but also in everyday life. Mediolateral balance is of particular interest, since it is more affected by ageing and disease and since its deterioration has been associated with an increased risk of falling (*Puszczałowska-Lizis, Bujas & Omorczyk, 2016*; *Melzer, Kurz & Oddsson, 2010*; *Hilliard et al., 2008*).

In light of the above, the aim of the present study was to investigate the effects of a single dose of contract-relax proprioceptive neuromuscular facilitation stretching of hip adductors and abductors on mediolateral dynamic balance.

## MATERIAL & METHODS

### Participants

The research protocol was approved by the Senate Commission for Ethics of Scientific Research at the University School of Physical Education in Wrocław. Criteria for exclusion were: impaired vision or hearing, central nervous system disorders, any balance impairment, lower extremity joint or sacroiliac dysfunction, or joint hypermobility syndrome. All included participants ($n = 45$, age 19–23 years) signed informed consent and they were assigned to the intervention group (IG) or the control group (CG). Participant characteristics are presented in Table 1.

### Procedures

All subjects visited the laboratory 24 h before testing session to undergo anthropometric measures and familiarize with the balance test and stretching exercises. They performed the balance test twice and then three repetitions of CRS (only intervention group). On the next day, in the testing session, the balance test was carried out before (Pre) and immediately after CRS in the intervention group or after 5-minute rest in sitting position in the control group (Post).

## Contract-relax proprioceptive neuromuscular facilitation stretching (CRS)

Stretching was administered to both limbs targeting the adductor and abductor muscles of the hip, in which the order of treatment was reversed in half of the sample (right and then left limb or vice versa). The CRS involved passive stretch (to the sense of discomfort) of the target muscles, then the 10-second of 50% maximal voluntary isometric contraction (MVIC) of the same muscles, followed by relaxation (5 s) and passive movement into further stretch (5 s). *Feland & Marin (2004)* found that submaximal contractions (20–60% MVIC) administered using the CRS method are just as effective as techniques that apply maximal contractions. There were performed three repetitions of the CRS on each muscle group.

## Dynamic balance assessment

Mediolateral dynamic balance was measured using the stabilometric platform (Libra; EasyTech, Salerno, Italy). The Libra platform is an electronic oscillating balance board that measures mediolateral tilt from $-15°$ to $+15°$ to an accuracy of $1°$. The platform was placed on a level surface using the manufacturer-supplied mat and a computer with dedicated software was connected to the device to register mediolateral tilt. All testing was performed in a quiet, well-lit, temperature-controlled room to minimize external influences. The participant stood barefoot on the balance board and was monitored by personnel for safety precautions in case of loss of balance. Feet were maintained parallel to one another and the upper limbs rested freely along the trunk. In addition, the participants kept their knees extended throughout the test in order to exclude the effects of the knee joint on stability. The task was to maintain balance on the device according to a pre-set level of deviation. During the trial, the participant observed a monitor that showed a sway line indicative of the amount of deflection from the center line (indicative of zero sway). Two borderlines paralleled the center line and marked the sway threshold (Fig. 1). If the participant did not maintain balance on the board, the sway line would deviate from the center line and an audio signal was sounded if one of the borderlines was crossed. The level of difficulty was set to maximum on the device permitting only minimal sway. Dynamic balance was quantified based on the amount of sway registered during the 30-s trial. The angular and temporal values of sway displacement were graphically represented on a stabilogram and used to extract stability outcomes:

- Total area of sway (TA)—the summed internal (within the threshold limit) and external (outside the threshold limit) area created by sway line amplitude that deviated from the center line [°s]
- External area of sway (EA)—the summed external area created by sway line amplitude outside the threshold limit [°s]
- External time (ET)—the summed time when sway line amplitude exceeded the threshold limit [s]
- Reaction time (RT)—the longest interval in which sway line amplitude exceeded the threshold limit [s]
- Global index (GI)—a weighted measure of the variables described above calculated on a relative 100-point scale in which the lower the score the better functional balance.

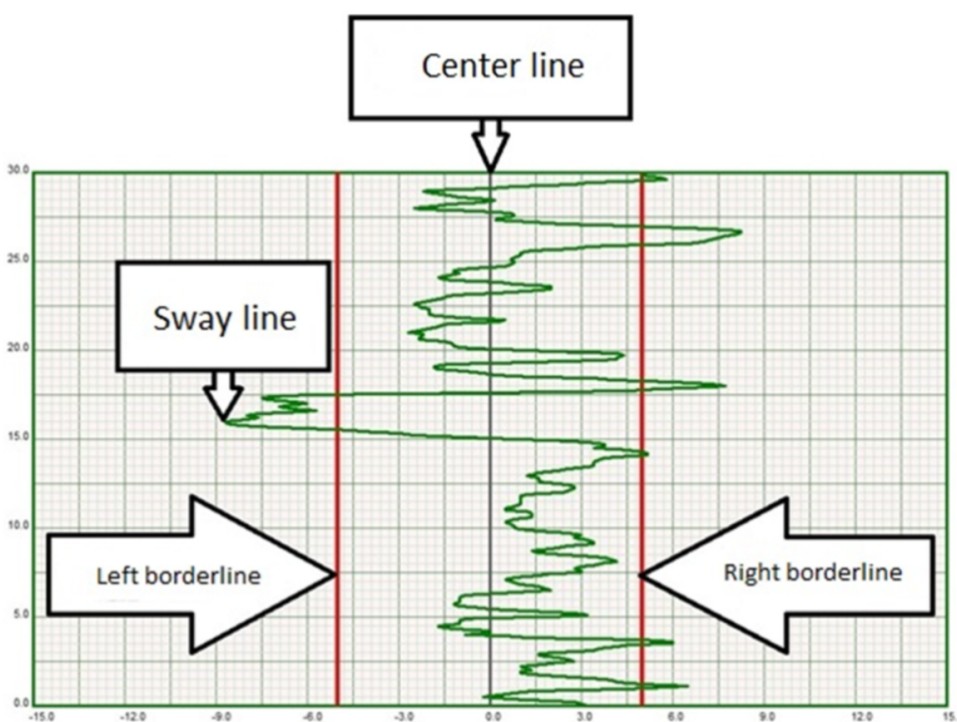

**Figure 1** Exemplary stabilogram illustrating sway line amplitude in relation to center and border lines.

## Statistical analysis

All data in the text and figures are presented as mean ± SD. Normal distribution of the data was examined using a Kolmogorov–Smirnov normality test. Initially, mixed-model ANOVA, repeated-measures analysis with one between factor (gender) and one within factor (time) was conducted in the intervention group (IG) to determine if there were differing responses because of gender. There was no significance found between genders, so the male and female scores were combined, and this group's performance was compared with that of the control group (CG) using mixed-model ANOVA, repeated-measures analysis with one between factor (group; IG and CG) and one within factor (time; Pre and Post). Bonferroni post-hoc multiple comparison was performed if a significant main effect was observed. For each ANOVA, partial eta-squared was calculated as measures of effect size. Values of 0.01, 0.06, and above 0.14 were considered as small, medium, and large, respectively. In the case of non-normality of experimental data (EA and ET), we applied alternative, non-parametric statistical analysis. In that case, a Wilcoxon signed-rank test was utilized to analyze relationships between pairs of related samples (time) and a Mann–Whitney $U$ test was used for analysis of pairs of independent samples (groups). Statistical analyses were performed using Statistica 13.1 software (Dell, Round Rock, TX, USA). The significance level was set at alpha = 0.05.

## RESULTS

### Global index

Mixed-model ANOVA indicated a significant main effect for groups ($F = 5.02$, $p = 0.03$, partial eta squared $\eta_p^2 = 0.11$, observed power = 0.6), time ($F = 6.96$, $p = 0.01$, $\eta_p^2 = 0.14$, observed power = 0.7) and the group x time interaction ($F = 5.74$, $p = 0.02$, $\eta_p^2 = 0.12$, observed power = 0.7). Bonferroni test revealed that GI in the intervention group was significantly lower after CRS (Pre $3.45 \pm 1.34$ vs. Post $2.42 \pm 0.73$; $p = 0.0001$). There was no significant difference between Pre and Post in the control group ($3.62 \pm 0.98$ vs. $3.57 \pm 1.38$; $p = 0.9$). IG values measured in Pre did not differ ($p = 0.7$) between IG and CG. IG values in Post condition were lower for IG than CG ($p = 0.0007$) (Fig. 2).

### Total area

Mixed-model ANOVA showed a significant main effect for groups ($F = 4.83$, $p = 0.03$, $\eta_p^2 = 0.10$, observed power = 0.6), time ($F = 7.23$, $p = 0.01$, $\eta_p^2 = 0.14$, observed power = 0.8) and the group $\times$ time interaction ($F = 6.72$, $p = 0.01$, $\eta_p^2 = 0.14$, observed power = 0.7). Bonferroni test established a decrease of TA in Post condition in the intervention group (Pre $69.18 \pm 16.45°$ s vs. Post $56.42 \pm 9.69°$ s; $p = 0.0001$) but not in the control group (Pre $71.16 \pm 12.96°$ s vs. Post $70.93 \pm 16.20°$ s; $p = 0.9$). There was no significant difference in TA between IG and CG in Pre ($p = 0.7$), however in Post the TA values were lower for IG than CG ($p = 0.0005$).

### External area

In the intervention group, a Wilcoxon test demonstrated a reduction of EA after the stretching procedure (Pre $4.61 \pm 5.49°$ s vs. Post $1.03 \pm 1.30°$ s; $p = 0.00004$). In the control group there was no significant difference between Pre and Post condition (Pre $5.86 \pm 5.94°$ s vs. Post $4.11 \pm 5.09°$ s; $p = 0.7$). Mann–Whitney test revealed that EA was significantly lower in IG as compared to CG ($p = 0.002$) only in Post but not in Pre ($p = 0.4$).

### External time

Wilcoxon test showed that ET decreased in the intervention group after stretching (Pre $2.57 \pm 1.97$ s vs. Post $1.06 \pm 1.15$ s; $p = 0.0004$); however, in the control group ET remained almost unchanged (Pre $2.71 \pm 1.64$ s vs. Post $2.71 \pm 2.12$ s; $p = 0.9$). Mann–Whitney test indicated that ET values measured in Pre did not differ ($p = 0.6$) between IG and CG. ET values in Post condition were lower for IG than CG ($p = 0.003$).

### Reaction time

Mixed-model ANOVA demonstrated a significant main effect for groups ($F = 10.29$, $p = 0.003$, $\eta_p^2 = 0.19$, observed power = 0.9) and the group $\times$ time interaction ($F = 5.69$, $p = 0.02$, $\eta_p^2 = 0.12$, observed power = 0.7). Bonferroni test revealed a decrease of RT in Post condition in the intervention group (Pre $1.14 \pm 0.68$ s vs. Post $0.53 \pm 0.49$ s; $p = 0.0003$) but not in the control group (Pre $1.33 \pm 0.74$ s vs. Post $1.44 \pm 1.09$ s; $p = 0.7$). There was no significant difference in RT between IG and CG in Pre ($p = 0.4$), however in Post condition the TA was lower for IG than CG ($p = 0.003$).

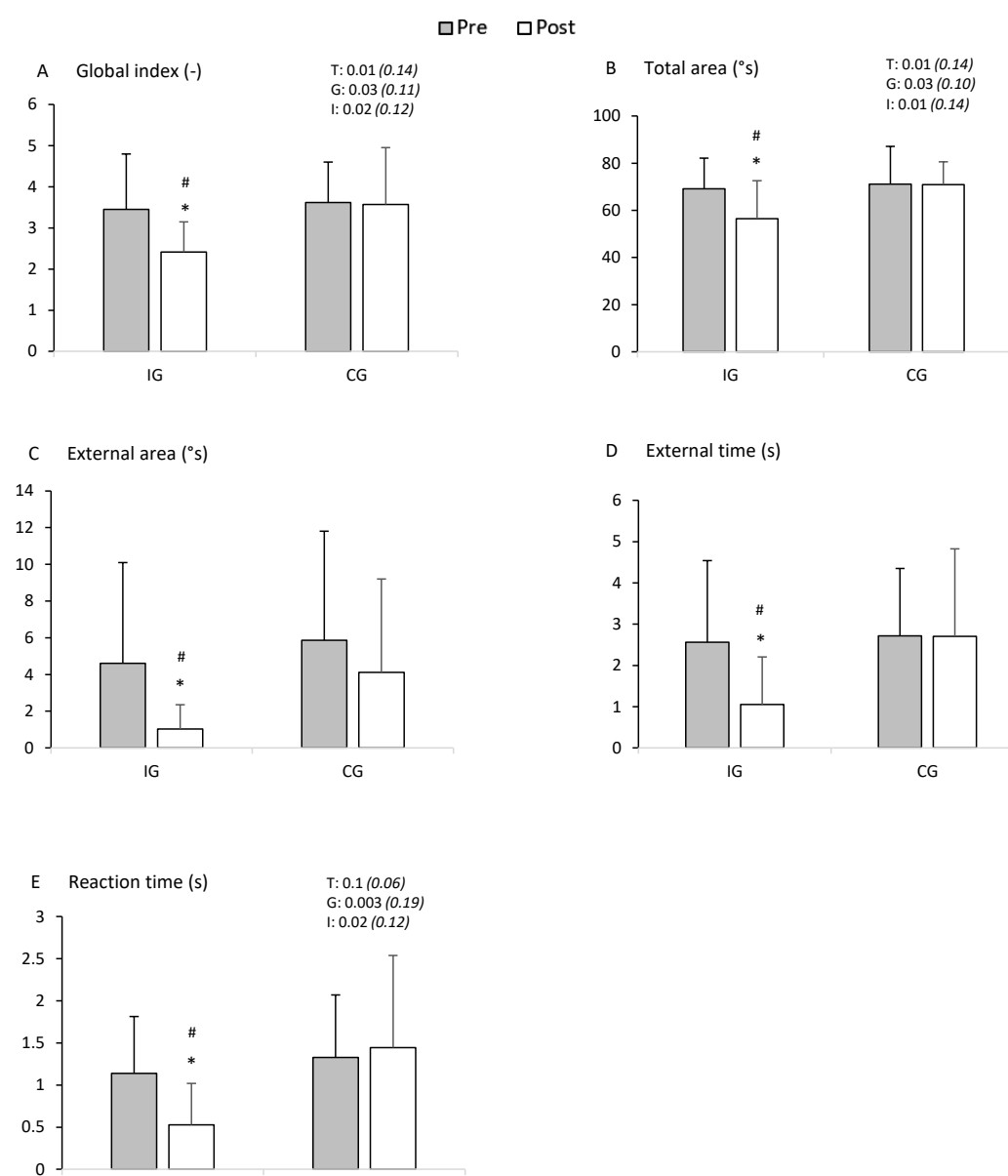

**Figure 2  Measurements of body balance before (Pre) and immediately after (Post) CRS in the intervention group (IG) or after 5-minute rest in the control group (CG).** (A) global index, (B) total area, (C) external area, (D) external time, (E) reaction time. T, time effects; G, group effects; I, interaction effects. ∗$P < 0.001$, significantly different from Pre. #$P < 0.01$, significantly different from CG.

## DISCUSSION

The aim of our study was to investigate the effects of a single dose of CRS of hip adductor and abductor muscles on mediolateral dynamic balance. Analysis of the results showed that single dose of CRS significantly improved mediolateral balance. The improvement of balance was manifested by the magnitude decrease in the postural sway variables, which

took place only in the intervention group, but not in the control group. In the intervention group we observed after stretching lower values of balance global index, sway area and reaction time. This proves that after CRS application, the subjects reacted faster to the support surface perturbations, which resulted in a decrease in a sway amplitude. In other words, the dynamic balance of the body was maintained more efficiently.

These results are in line with those reported by *Ryan, Rossi & Lopez (2010)*, who investigated the effects of CRAC intervention targeting the hamstrings, quadriceps, iliopsoas and plantar flexor muscles on balance. In their study, each stretch consisted of a passive initial stretch to the point of mild tension or restriction, followed by a 7-second isometric contraction of the target muscle. Then a concentric contraction of the opposite muscle group was performed for 4 s. The procedure was conducted four times. The authors found that CRAC method improved mediolateral balance. They indicate the neurological facilitation from the contract–relax stage, or irradiation overflow from the antagonist-contract phase, as potential mechanisms leading to the improvement of body balance. The results of our study, in which the CRS method was used, seem to confirm the first mechanism concerning the neurological facilitation. *Young & Elliot (2001)* argue that a neurological facilitation mechanism may result in a lingering activation of motor units. Furthermore, *Ostering et al. (1990)* state that the CRS technique leads to increased electromyographic (EMG) activity through the isometric contraction, which may help to counterbalance the tendon slack associated with acute static stretching.

*Ghram, Damak & Costa (2017)* came to completely different conclusions. They stated that CRS of the quadriceps, hamstrings, anterior tibialis, and calf muscles impaired static balance control. In their study, the subjects performed a maximal voluntary isometric contraction (MVIC) of the target muscle during 5-second, followed by 5-second of relaxation and 5-second of static stretching. We used twice as long (10 s) isometric contraction time, but with 50% MVIC. Both the difference in time of isometric contraction and its strength could influence the achievement of such different results.

This study had some limitations. First of all, we did not measure the changes in muscle tone or range of motion. Supplementing the experiment with measurement of physiological properties of the stretched muscles, would allow for a more detailed interpretation of the results. Secondly, we used a healthy and young population, and our results are not generalizable to other populations. Thirdly, we measured only acute effects of CRS so we do not know how long the improvement in balance can be maintained.

All the limitations do not negate the fact that our study is of practical importance because it closely simulates the type of interventions used by fitness and rehabilitation professionals. Our results suggest that contract-relax proprioceptive neuromuscular facilitation stretching can be used as one of the means to improve or counteract impairments of body balance.

Future research should include the elderly, patients with limited joint range of motion (for whom stretching is particularly indicated), and patients with impaired the visual, vestibular, and somatosensory systems. Futures studies should also include stiffness and electromyographic analyses of the stretched muscles and should evaluate how long the effect is maintained.

## CONCLUSIONS

A single dose of contract-relax proprioceptive neuromuscular facilitation stretching of the hip adductor and abductor muscles improved mediolateral dynamic balance.

### Funding

The authors received no funding for this work.

### Competing Interests

The authors declare there are no competing interests.

### Author Contributions

- Rafał Szafraniec conceived and designed the experiments, performed the experiments, analyzed the data, authored or reviewed drafts of the paper, approved the final draft.
- Krystyna Chromik contributed reagents/materials/analysis tools, prepared figures and/or tables, authored or reviewed drafts of the paper, approved the final draft.
- Amanda Poborska performed the experiments, analyzed the data.
- Adam Kawczyński authored or reviewed drafts of the paper, approved the final draft.

### Human Ethics

The following information was supplied relating to ethical approvals (i.e., approving body and any reference numbers):

The research protocol was approved by the Senate Commission for Ethics of Scientific Research at the University School of Physical Education in Wrocław.

### Data Availability

The raw data are provided in Supplemental Information 1.

### Supplemental Information

Supplemental information for this article can be found online at http://dx.doi.org/10.7717/peerj.6108#supplemental-information.

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
