# Peer review of "Acute effects of contract-relax proprioceptive neuromuscular facilitation stretching of hip abductors and adductors on dynamic balance"

_PeerJ, doi:10.7717/peerj.6108_

## Round 0.1 · original submission · Major Revisions

Although the study is of interest, the reviewers have raised many questions and concerns that need to be addressed in the revised manuscript. In particular, the focus of the Introduction and Discussion needs to be improved to provide a better rationale of the specific intervention, consider using an omnibus test and correct for multiple comparisons.

Reviewer 1 ·

Basic reporting

This is an original article. The article presents significance in the content, however, the description of it needs to be improved. The aim of the present study was to investigate the effects of a single dose of contract-relax proprioceptive neuromuscular facilitation stretching of hip adductors and abductors on dynamic balance in the frontal plane.

Experimental design

Research question well defined.
Methods described with sufficient details.
Statistical analysis was simplified and unspecific to this study.

Validity of the findings

The study presents interesting results, but with a limited analysis.

Additional comments

Title:
I recommend changing “frontal plane” to “medio-lateral balance”. And avoid writing your title as a question.
Abstract:
Please, add your main purpose (aim) in the abstract.
Add a rationale for your hypothesis.
Please, follow a single order (i.e. p=000.x; and d=0.0x or vice versa).
Introduction:
This section is too long and present unnecessary and basic information about stretching and postural control. Your introduction needs to be more specific.
Now, you added the aim, but forgot the hypothesis/rationale.
Methods:
Participants: Did you define the power of this study?
You have different genders in your same sample. Could you explain how anthropometric differences (i.e. pelvic width) can affect the balance?
Please, add the # of the IRB.
Procedures: Your purpose does not require “no feedback” and “visual feedback” conditions. Why did you do both?
Did you measure the intensity of the stretching bout (i.e. Level of discomfort)?
Please, add the version/model of the Libra stabilometric platform.
“To compensate for learning effects, the order of the trials was reversed in half of the sample (crossover design). Is this a crossover design? Explain.
Statistical Analysis: Why did you use a t-test if you have 3 factors (leg [right and left]; pre- and post-CR results; and with and without feedback)?
You have to describe your index to the Cohen’s d in this section.
Please, review your statistical analysis.
Results:
You cannot use this description of the significance: p = 0.000. Change to p < 0.001. This section is confusing, and based on effect sizes.
It is not possible to trust in your results, after your Statistical Analysis section.
Discussion:
This section is too long and present unnecessary information. Your introduction needs to be more specific, and related to your results.
Conclusion:
Please, improve your conclusion, and follow your purpose.

·

Basic reporting

This is an interesting study, and the effects of stretching on dynamic balance have practical relevance to fitness and rehabilitation interventions. However, the Introduction does not provide enough background and rationale for why it is expected that CR stretching would affect dynamic balance. The authors focus primarily on what balance is, what sensory systems play a role, etc. in the Introduction and very little on the mechanisms of stretching and how/where/why that would affect the control of balance. Although a background on balance is important, the authors present some information that is irrelevant and never touched upon again in the context of the current study (e.g., why do I care that there are two motor strategies [ankle and hip] to maintain balance, and how does this relate to the current study?). More emphasis needs to be placed on stretching (particularly CR stretching), how it works, how it changes the system, and the mechanisms by which this would alter the control of balance.

The authors include a lot of background research in this manuscript in both the introduction and discussion sections. However, there are many long paragraphs in which several ideas are buried, and it is difficult to know what are the main points. For example, the first paragraph in the Discussion (lines 186-219) seems to present a large number of other studies that examine stretching and balance control yet I have no idea whether I am supposed to take away that the current study is in agreement or disagreement with these prior studies. In contrast, the next paragraph of the discussion section (Lines 220-222) is only two sentences and does not provide enough detail. The organization of the Introduction and Discussion sections need to be revisited in order to enhance the readability of the manuscripts and better present both background ideas and the results of the current study.

Raw Data – The data presented in the supplemental file is well-organized and easy to follow. However, the only data given are the calculated variables (TA, EA, ET, RT, and GI) and not the underlying raw data used to get those numbers.

Experimental design

The research question is well-defined and the methods are generally described in sufficient detail and with enough information to replicate. However, I have several concerns regarding the experimental design:

1. Learning effects – The authors have already taken into account some aspects of learning effects by reversing the order of trials (feedback vs. non-feedback trials) in half of the participants. However, what the authors have no accounted for is the learning effects that may occur by simply repeating the balance task a total of four times without having performed stretching in the middle. To account for such learning effects the authors should include a control group that does not receive stretching.

2. Statistical tests – It is good that the authors tested normality of the data before selecting a parametric statistical test and including the effect sizes is a huge plus. The actual statistical tests used (dependent samples/paired t-tests) may not be the most appropriate. The authors are comparing 9 different metrics across two different conditions. Some sort of ANOVA may be more appropriate to account for the multiple comparisons and allow better comparisons between what happens in with feedback and without feedback conditions.

Validity of the findings

Given the problems stated above with the experimental design, I am not yet convinced that the results support the conclusions. Particularly because many of the p-values (especially for without feedback condition) are barely below the alpha of 0.05. Although the conclusions will likely hold (given the high effect sized with visual feedback, for example), the authors need to re-evaluate their results and conclusions after addressing the experimental design deficiencies.

The results need to be placed in better context with prior studies; addressing the prior comment regarding long, unorganized paragraphs will help place them in context.

Additional comments

No comments.

---

## Round 0.2 · Major Revisions

Although the revised manuscript has much improved, it does not meet the criteria for publication in PeerJ. The authors should carefully revise and restructure the introduction and discussion to make it more concise and clarify the rationale and primary findings of the study.

·

Basic reporting

The major remaining critique of this manuscript is the organization of the Introduction and Discussion sections. Although the Introduction has been modified to address prior critiques (e.g., irrelevant information removed, and more discussion of mechanisms of stretching), it remains hard to follow. Instead of having one long paragraph, the authors should split the Introduction into several smaller paragraphs, each focused on a main point that helps build the rationale for the study. The Discussion section is similarly plagued by a long paragraph in which it is hard to identify the main points. Reorganizing both the Introduction and Discussion sections is necessary to enhance the readability of the manuscripts and better present both background ideas/rationale and the results of the current study.

Experimental design

The resubmitted manuscript is much improved by inclusion of a control group and adjusted statistical analyses.

Validity of the findings

No comment.

---

## Round 0.3 · Minor Revisions

The manuscript will be accepted pending some final minor changes.

·

Basic reporting

The Introduction and Discussion sections of the manuscript are much improved. I suggest the following minor edits:

- Line 45: Replace “Balance control have” with “Balance control has”
- Line 56: Replace “The changes in the muscle-tendon unit stiffness” with “Changes in muscle-tendon stiffness”
- Line 63: Replace “Two of the PNF techniques” with “Two PNF techniques”
- Line 83: I don’t think Versteeg et al. 2016 addresses ML balance (as it is an AP model of balance control).

Experimental design

The research question is well-defined and the methods are generally described in sufficient detail and with enough information to replicate.

Validity of the findings

No comment

---

## Round 0.4 · accepted · Accept

The authors adequately addressed the final comments.